# SDR-Implemented Passive Bistatic SAR System Using Sentinel-1 Signal and Its Experiment Results

**Weike Feng** [1] , **Jean-Michel Friedt** [2,*] and **Pengcheng Wan** [1]

1 Early Warning and Detection Department, Air Force Engineering University, Xi'an 710051, China; fengweike007@163.com (W.F.); wppcheng@163.com (P.W.)
2 Time & Frequency Department, FEMTO-ST, 25030 Besancon, France
* Correspondence: jmfriedt@femto-st.fr

**Abstract:** A fixed-receiver mobile-transmitter passive bistatic synthetic aperture radar (MF-PB-SAR) system, which uses the Sentinel-1 SAR satellite as its non-cooperative emitting source, has been developed by using embedded software-defined radio (SDR) hardware for high-resolution imaging of the targets in a local area in this study. Firstly, Sentinel-1 and the designed system are introduced. Then, signal model, signal pre-processing methods, and effective target imaging methods are presented. At last, various experiment results of target imaging obtained at different locations are shown to validate the developed system and the proposed methods. It was found that targets in a range of several kilometers can be well imaged.

**Keywords:** passive bistatic radar; synthetic aperture radar; high-resolution imaging; Sentinel-1; software-defined radio





## 1. Introduction

Passive bistatic radar (PBR), which uses non-cooperative emitting sources for target illumination, has attracted increasing attentions in the last decades, owing to its unique remote sensing capabilities [1]: (1) no need for frequency allocation; (2) no pollution to the crowded radio frequency environment; (3) working well without using self-designed radar transmitter; and (4) obtaining the target scattering information from a particular bistatic angle. Along with its development, various applications have been realized by PBR, including moving target detection and range-Doppler mapping, synthetic aperture radar (SAR) imaging, air/sea target inverse SAR (ISAR) imaging, displacement estimation, and coherent change detection [2–7].

Among these applications, passive bistatic SAR (PB-SAR) imaging has been used for providing abundant information (e.g., size, shape, and scattering intensity) of stationary targets. Depending on the type of emitting source, the majority of PB-SAR systems can be divided into two categories: fixed-transmitter mobile-receiver PB-SAR (FM-PB-SAR) and mobile-transmitter fixed-receiver PB-SAR (MF-PB-SAR). In FM-PB-SAR, the emitting source is stationary while the receiver is mobile to obtain a high cross-range resolution. Different non-cooperative signals with a certain bandwidth can be used for FM-PB-SAR [8–11], such as broadcasting digital TV signal (either terrestrial or satellite) and communication Wi-Fi signal. If properly designed, the FM-PB-SAR systems can achieve well-focused image of the targets in a local area. However, in spite of the range resolution determined by the signal bandwidth, the cross-range resolution of FM-PB-SAR is closely related to the receiver motion, which may be restricted in practical implementations (such as the rail length in [8,11]). In MF-PB-SAR, the employed non-cooperative source is mobile, while the receiver is fixed on the ground; thus, the system can be more convenient to implement than FM-PB-SAR to obtain a high cross-range resolution (if no transmitter-receiver synchronization is considered). For example, TerraSAR-X was used for MF-PB-SAR as the non-cooperative emitting source in [12,13], and GNSS signal was applied for target imaging in [14,15].

In recent years, advanced C-band SAR satellites, Sentinel-1 A/B, launched by European Space Agency (ESA) in 2014/2016, have become a burgeoning focus for MF-PB-SAR studies. For example, the COBIS system is presented in [16,17], demonstrating the feasibility of Sentinel-1 signal for MF-PB-SAR imaging. In these works, with specially designed receiver and accessories, timing, frequency, and position synchronization between satellite transmitter and ground receiver were conducted and the back projection (BP) imaging algorithm was employed to obtain the image of targets. Moreover, via the Terrain Observation with Progressive Scans SAR (TOPSAR) technique, Sentinel-1 can cover three different sub-swaths with a scanning beam; thus, the pulse repetition intervals (PRIs) and amplitudes of the received signal pulses change in a long data-receiving period. In such a case, to improve the cross-range resolution, a multiple-aperture focusing method based on the autoregressive (AR) model is proposed in [18], and a compressive sensing (CS)-based azimuth profile reconstruction method is proposed in [19].

Although having been well validated, there are still some problems of current Sentinel-1-based MF-PB-SAR studies: (1) the specialized and exquisite ground receiver makes the system expensive and difficult to implement; (2) the requirement of exact satellite information needs accurate and complicated transmitter–receiver synchronization, reducing the system flexibility; (3) existing target imaging methods, such as the BP- and CS-based methods presented in [17,19], induce high computational costs. Recently, software-defined radio (SDR) hardware has been applied to radar applications with an increasing interest, such as the multichannel digital weather radar in [20], the high-range-resolution radar in [21], and the ground-based multiple-input multiple-output (MIMO) radar in [22], which shows a low-cost and flexible solution for MF-PB-SAR development because of its reconfigurability and structural universality. Therefore, in this study, to reduce the development cost and improve the system flexibility, we demonstrate the use of commercial-off-the-shelf (COTS) SDR hardware to implement an MF-PB-SAR system using Sentinel-1 as its emitting source, making the system handheld and deployable on remote field areas. Moreover, without the requirement of exact satellite information, an effective approach and its high-resolution version realized based on efficient 2D CS algorithm are proposed for target imaging with reduced computing complexity.

The remainder of this paper is organized as follows: In Section 2, Sentinel-1 and the designed MF-PB-SAR system are introduced. In Section 3, the signal model, signal pre-processing methods, and target imaging methods are presented. In Section 4, various experiment results of target imaging at different locations are presented to show the effectiveness of the developed system and the proposed methods. Finally, Section 5 concludes this paper.

## 2. Sentinel-1 and System Overview

Sentinel-1 is a set of two SAR satellites at an altitude of 693 km launched by ESA to provide continuous, all-weather, and day-and-night imagery of the Earth surface at C-band. Its resolution, area coverage, and revisit time are more advanced than some others, such as ERS-1/2 and ENVISAT ASAR. The satellites are in a near-polar and sun-synchronous orbit, sharing the same orbit plane with a 180° orbital phasing difference. The repeat cycle of each satellite is 12 days; therefore, the maximal repeat cycle at the equator is 6 days, and the revisit rate is greater at higher latitudes, e.g., the Sentinel-1 A/B can be observed at least every three days above 45° N latitude and twice each day at Arctic latitudes. As opposed to dedicated observations as planned for RADARSat or TerraSAR, the continuous monitoring of the Earth by Sentinel-1 allows for accurately predicting its flight time from its orbit period and the observations published on the ESA Copernicus website [23] with a sub-minute resolution. Knowing the ground station location, its flight geometry can be collected from orbital parameter computation, in our case obtained from the Heavens Above website [24].

To obtain a large swath width and a moderate resolution at the same time, the interferometric wide (IW) swath mode is used as the default data acquisition mode of Sentinel-1 when scanning the land by using the TOPSAR technique. In IW mode, the satellite transmitted signal is a set of chirp pulses with three different PRIs corresponding to three overlapped sub-swaths on the land, as shown in Figure 1. In such a case, since the satellite moves and periodically steers its antenna beam from sub-swath 1 to sub-swath 3, the PRI and amplitude of the IW signal received by a directional antenna fixed on the ground changes along with time, as shown in Figure 2. It can be seen that three IW signals have different PRIs, i.e., 582.37 µs (IW 1), 688.88 µs (IW 2), and, 593.18 µs (IW 3), as identified from the signal autocorrelation or decoding the level-0 raw data provided by ESA on the Copernicus website. Furthermore, due to the gains of the satellite antenna and the ground receiving antenna, the amplitudes of different IW signals change greatly, and the signal-to-noise ratio (SNR) is relatively low in some parts. Over sea or remote areas, such as the North Pole, larger swath width and lower resolution in the so-called extra wide (EW) measurement mode are used by Sentinel-1 with different PRIs from the IW measurement mode. The developed system can receive either IW or EW signal by simply changing its corresponding software parameters, making the implementation of the system feasible over most ground covered areas of the Earth.

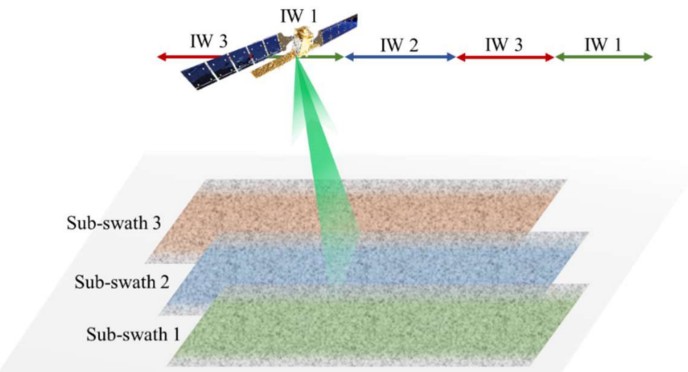

**Figure 1.** The scanning process in the IW mode of Sentinel-1 using the TOPSAR technology, where each sub-swath corresponds to individual IW signal, designated by IW 1, IW 2, and IW 3.

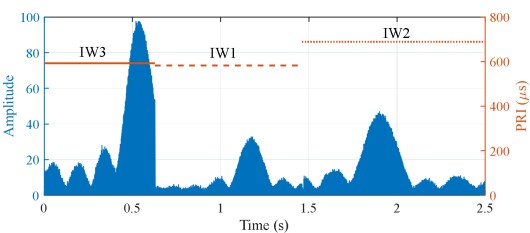

**Figure 2.** The Sentinel-1 signal received by a directional antenna fixed on the ground: three different PRIs and amplitude differences of these IW signals can be observed.

The center frequency and bandwidth of the Sentinel-1 signal are 5.405 GHz, corresponding to a wavelength of about 5.55 cm, and up to 100 MHz programmable, respectively. In this study, the bandwidth of the system received signal is limited by the adopted SDR receiver. Considering high-precision data acquisition and transmission in the dual-channels of the SDR receiver, the upper signal bandwidth is 30 MHz, providing a bistatic range resolution of 10 m, as shown in Figure 3, where Figure 3a shows the real part of a received IW 3 signal pulse and Figure 3b shows its corresponding range ambiguity function.

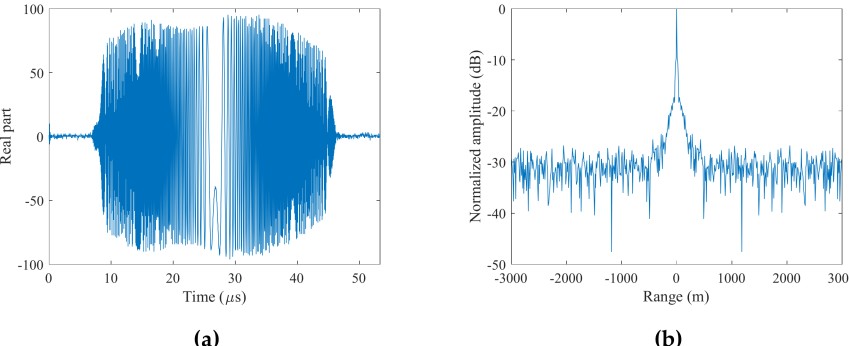

**(a)**                    **(b)**

**Figure 3.** (**a**) Real part of a received signal pulse (IW 3) and (**b**) its range ambiguity function, showing a bistatic range resolution of 10 m with the SDR receiver setup limited to a 30 MHz bandwidth.

As shown in Figure 4, the SDR implemented MF-PB-SAR system is mainly composed by two helical antennas (one is used for directly receiving the Sentinel-1 signal, acting as the reference, and another is used for receiving the target reflections in the observed scene, named as surveillance), an SDR receiver (Ettus B210), a single-board Raspberry Pi 4 (RPi 4) used to collect and store the data, and a host computer for signal processing and result displaying. Detailed information of these components is given as follows.

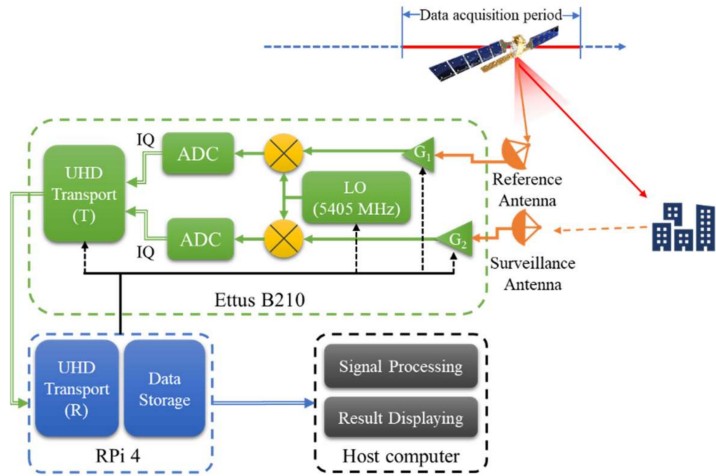

**Figure 4.** The general structure of the developed MF-PB-SAR system, including one SDR receiver (Ettus B210), a host computer, a RPi 4, and two antennas (reference and surveillance antennas).

(1)    Antennas. The helical antennas are designed according to [25] by winding four turns of enameled copper wire around a 15 mm hollow Teflon tube, providing a frequency band from 4.8 to 8.5 GHz, including the Sentinel-1 signal band. The low antenna gain design of 5 dB is selected for a wide beam to observe a broad imaging scene. The reference antenna is set at an angle matching the satellite elevation during its pass, and the surveillance antenna is set pointing to the targets in the observation scene.

(2)    SDR Receiver. The 70 MHz–6 GHz COTS Ettus B210 dual-channel SDR hardware is chosen as the receiver of the developed MF-PB-SAR system due to its high performance and low cost. Two receiving channels share a local oscillator (LO), setting at 5405 MHz, and the parameters (e.g., gain and offset) of each channel are adjusted according to the practical condition. As mentioned above, the data sampling frequency of Ettus B210 is set at 30 MS/s considering the dual-channel high-precision data acquisition and transmission.

(3)    RPi 4. The single-board computer RPi 4 fitted with a USB 3.0 port collects the data from the B210 and stores them in its RAM. Data collection duration and communication speed are maximized by reducing the sample resolution to a single byte/sample.

Hence, at a rate of 30 MS/s, the complex dual-channel sample requires 7.2 GB/minute. The 8 GB RAM version of the RPi4 can hence hold 1 min worth of data or twice the pass duration of the satellite, allowing sample flexibility in data acquisition starting time.

(4) Host computer. Once the 7.2 GB of data have been collected in RPi 4, they are transferred to the host computer. Signal pre-process and process, as are introduced in the next Section, are then conducted, and the target imaging results are finally displayed.

The overall working flow of the developed MF-PB-SAR system can be summarized as:

(1) The reference antenna orientation is set to ensure maximum reception of the direct satellite signal on the foundation of its relative position to Sentinel-1, where the scheduled pass geometry of the satellite is queried in public information, such as the Heavens Above website. The surveillance antenna is set to face the targets, and, to reduce the direct-path interference (DPI) in the surveillance channel, it is placed properly to make the satellite within its side lobe during the measurement.

(2) The time window for data acquisition is determined from past datasets made available on the ESA Copernicus website, whose file name includes the beginning and ending of the data sampling with one second resolution. While a horizon-to-horizon satellite pass lasts 9 min, a given area is only illuminated for a few seconds. As introduced before, the satellite repeats its pattern over a period of 12 days. By searching the pattern over the scheduled site in the previous public raw data, the future accurate acquisition time within several seconds can be determined.

(3) The signal of two channels is collected and converted into complex format by Ettus B210 and transmitted to RPi 4. RPi 4 stores the data in its RAM and later transfers them to the host computer, which finally analyzes and processes the data with the proposed methods.

With a current consumption of 1.55 A at 5 V supply when running at full speed in performance mode (1.5 GHz) as needed to collect and store the data, the setup allows for collecting up to 85 datasets autonomously on a single 2200 mAh battery pack as used in the system.

## 3. Signal Model and Processing Methods

### 3.1. Signal Modeling and Pre-Processing

In this subsection, the signal modeling and pre-processing methods for the developed MF-PB-SAR system are presented.

Firstly, assume the signals received by the reference and surveillance antennas (as shown in Figure 4) are defined as reference signal and surveillance signal and expressed as $s_{ref}(n)$ and $s_{sur}(n)$ in the discrete time domain, respectively. Here, $n = f_s t$, $t$ denotes time, and $f_s$ denotes the system sampling frequency. Since the SNR of the received signal changes with time, the first signal pre-processing step we conducted is to select a specific IW (EW) signal fragment and then determine its corresponding PRI for the following process. Here, we select the IW (EW) signal fragment sampled when the satellite just flies above and its beam just steers to the experiment site. As shown in Figure 2, the corresponding IW (EW) signal fragment can be selected according to the amplitude of the received reference signal. If the satellite is above the experiment site, the SNR of the reference signal is high. Thus, given an amplitude threshold $\chi$, set according to practical conditions, the starting and ending points of the selected IW (EW) signal fragment can be obtained as $n_s$ and $n_e$ with $s_{ref}(n_s) \geq \chi$ and $s_{ref}(n_e) \geq \chi$. We note that, out of the collected data samples, the selected IW (EW) signal fragment typically lasts a few hundred milliseconds at most.

For the selected signal fragment, a 2D reference signal matrix can be formed by

$$S_{ref} = [s_{ref}^1, s_{ref}^2, \ldots, s_{ref}^P] \in C^{N_r \times P} \tag{1}$$

where $N_r$ denotes the number of selected sampling points in each pulse, determining the system maximal detection range, $P$ denotes the number of selected pulses, determined by $n_s$ and $n_e$ as $P = \lfloor (n_e - n_s)/N_0 \rfloor$, and $s_{ref}^p (p = 1, 2, \ldots, P)$ can be expressed as

$$s_{ref}^p = [s_{ref}(n_s + (p-1)N_0), s_{ref}(n_s + (p-1)N_0 + 1), \ldots, s_{ref}(n_s + (p-1)N_0 + N_r - 1)]^T \in C^{N_r \times 1} \quad (2)$$

with $(\cdot)^T$ as matrix transpose, $N_0 = \lfloor f_s T \rfloor \geq N_r$ as the number of sampling points corresponding to the PRI $T$, depended on the IW or EW, and $\lfloor \cdot \rfloor$ as the floor function.

Only when the correct PRI is adopted for the selected signal fragment can the signal matrix in (1) be used for following process. Moreover, to ensure the signal matrix in (1) contains all samples of each IW (EW) signal pulse, the starting and ending points $n_s$ and $n_e$ should be fine-tuned after the initial detection. For example, for the reference signal shown in Figure 2 with an amplitude threshold of $\chi = 60$, the obtained signal matrix is presented in Figure 5 with the number of sampling points in each pulse as $N_r = 2401$, corresponding to a maximal detection range of 24 km given $f_s = 30$ MS/s. It can be seen from Figure 5a that the IW 3 signal fragment with a PRI of 593.18 μs is selected with $P = 251$ pulses, and, in order to contain all the samples of each pulse, some reserved sample points are included at the beginning of each pulse. If an incorrect PRI is used, the signal matrix is not well aligned, as shown in Figure 5b,c.

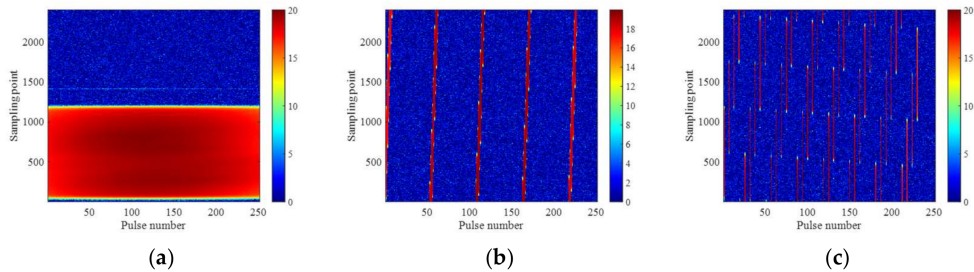

**Figure 5.** The obtained 2D reference signal matrices with (**a**) PRI = 593.18 μs (IW 3), (**b**) PRI = 582.37 μs (IW 1), and (**c**) PRI = 688.88 μs (IW 2).

Similarly, a 2D surveillance signal matrix can also be generated, expressed as

$$S_{sur} = [s_{sur}^1, s_{sur}^2, \ldots, s_{sur}^P] \in C^{N_r \times P} \quad (3)$$

where

$$s_{sur}^p = [s_{sur}(n_s + (p-1)N_0), s_{sur}(n_s + (p-1)N_0 + 1), \ldots, s_{sur}(n_s + (p-1)N_0 + N_r - 1)]^T \in C^{N_r \times 1} \quad (4)$$

In this study, as shown in Figure 6, we assume the ground target is located at $(x, y)$ with the height of 0, the reference and surveillance antennas are collocated at $(0, 0, 0)$, and the satellite is at $(-H \cot \varphi, y_{sat}^p, H)$ with $H$ as its height (693 km) and $\varphi$ as its incident angle (depending on the satellite beam) for the $p$-th pulse in the selected IW (EW) signal fragment (i.e., the $p$-th columns in (1) and (3)). To make it simple, we assume the satellite moves horizontally during the short data acquisition period, i.e., the satellite moves along the $y$ axis, as indicated by the red dotted line in Figure 6.

For the imaging process, a plane (i.e., $\alpha$-$o$-$\beta$) is formed by the satellite moving trajectory and the antenna position, as indicated by the yellow plane in Figure 6. On the $\alpha$-$o$-$\beta$ plane, the antennas are at $(0, 0)$, the target is at $(\alpha, \beta)$, and the satellite position corresponding to the $p$-th signal pulse is $(\alpha_{sat}^p, \beta_{sat})$ with $\alpha_{sat}^p = y_{sat}^p$ and $\beta_{sat} = -H / \sin \varphi$, where $\alpha_{sat}^p = \alpha_{sat}^c + [p - 1 - (P-1)/2]d$, $d = vT$, and $v$ denotes the linear speed of the satellite, which is estimated to be 7.49 km/s based on the satellite altitude and the orbit period. Since the SNR of the reference signal is the highest when the satellite is just above the antennas, the center satellite position is the same with the antenna, i.e., $\alpha_{sat}^c = 0$, if the IW (EW) signal fragment is ideally selected.

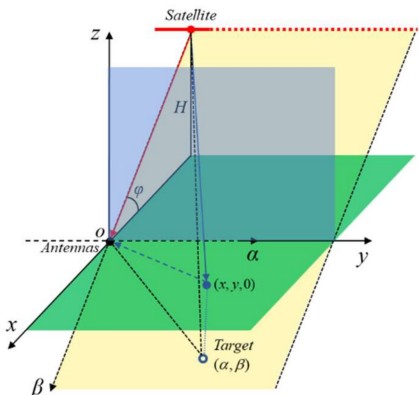

**Figure 6.** Imaging geometry of the developed SDR MF-PB-SAR system with the satellite height $H = 693$ km and the incident angle $\varphi$ of the satellite obtained from the Heavens Above website. The target in the $x$-$o$-$y$ plane is projected on the $\alpha$-$o$-$\beta$ plane for imaging process.

Assuming the satellite transmitted signal at the $p$-th pulse in the selected IW (EW) signal fragment is $g_0^p(\tau)$ in the continuous time domain with $\tau \in [0, T]$, the received signals by the reference and surveillance antennas can be expressed as

$$g_{ref}^p(\tau) = A^p g_0^p(\tau - \tau_{ref}^p), g_{sur}^p(\tau) = \sigma(\alpha, \beta) g_0^p[\tau - \tau_{sat}^p(\alpha, \beta) - \tau_{sur}(\alpha, \beta)] + g_{dpi}^p(\tau) \quad (5)$$

where $A^p$ denotes the amplitude of the reference signal in the $p$-th pulse, affected by the gains of satellite and reference antennas, $\sigma(\alpha, \beta)$ denotes the target reflection coefficient, $g_{dpi}^p(\tau)$ denotes the DPI signal (i.e., the satellite signal received directly by the surveillance antenna), and

$$\tau_{ref}^p = \sqrt{\left(\alpha_{sat}^p\right)^2 + \beta_{sat}^2}/c, \ \tau_{sur}(\alpha, \beta) = \sqrt{\alpha^2 + \beta^2}/c, \ \tau_{sat}^p(\alpha, \beta) = \sqrt{\left(\alpha_{sat}^p - \alpha\right)^2 + (\beta_{sat} - \beta)^2}/c \quad (6)$$

denote the delays from the satellite to the reference antenna, from the target to the surveillance antenna, and from the satellite to the target, with $c$ as the speed of light.

Based on (5) and according to the signal matrix formation procedure, the $n_r$-th $(n_r = 1, 2, \ldots, N_r)$ elements of $s_{ref}^p$ and $s_{sur}^p$ can be expressed as

$$s_{ref}^p(n_r) = g_{ref}^p\left\{\tau = (n_r - 1 - n^p)/f_s + \tau_{ref}^p\right\}, \ s_{sur}^p(n_r) = g_{sur}^p\left\{\tau = (n_r - 1 - n^p)/f_s + \tau_{ref}^p\right\} \quad (7)$$

where $n^p$ denotes the number of reserved sampling points as shown in Figure 5a.

In practice, the DPI component is always stronger than the target reflections and thus may make some targets undetectable in the imaging result. To solve this problem, the second signal pre-processing step we conducted is to suppress the DPI while keeping the target reflections. In this study, the LS-based method [11] is used to do so, giving

$$s_{sur}^p \leftarrow s_{sur}^p - U_p(U_p^{\mathrm{H}} U_p)^{-1} U_p^{\mathrm{H}} s_{sur}^p \quad (8)$$

where $(\cdot)^{\mathrm{H}}$ denotes conjugate transpose, $(\cdot)^{-1}$ denotes matrix inverse, and $U_p$ is constructed by the delayed copies of $s_{ref}^p$, expressed as

$$U_p = [u_p^1, u_p^2, \ldots, u_p^L] \in C^{N_r \times L} \quad (9)$$

where

$$u_p^l = [s_{ref}(n_s + (p-1)N_0 - l), \ldots, s_{ref}(n_s + (p-1)N_0 + N_r - 1 - l)]^T \in C^{N_r \times 1} \quad (10)$$

with $l = 1, 2, \ldots, L$ as the discrete time delays used to properly model the DPI.

Furthermore, in order to reduce the influence of amplitude differences of the signals in different pulses on target imaging, the last signal pre-processing we conducted in this study is to proportionally estimate $A = [A^1, A^2, \ldots, A^P] \in R^{1 \times P}$, based on which the received reference signals are compensated. It should be noted that, because the target reflection signals are much smaller and thus more affected by noise, amplitude differences of the surveillance signals in different pulses are not compensated in order to avoid increasing the noise level.

To proportionally estimate $A$, the amplitude maximum of the reference signal in each pulse is calculated first to obtain

$$|S_{ref}|_{\max} = [|s_{ref}^1|_{\max}, |s_{ref}^2|_{\max}, \ldots, |s_{ref}^P|_{\max}] \in R^{1 \times P} \tag{11}$$

Then, a cubic polynomial curve fitting method is used to obtain the estimation of $\widetilde{A}^p$, expressed as

$$|S_{ref}|_{\max}(p) \simeq \sum_{i=1}^{3} a_i p^i \rightarrow \widetilde{A}^p = \sum_{i=1}^{3} a_i p^i \tag{12}$$

At last, the received reference signals can be compensated by

$$S_{ref} \leftarrow [s_{ref}^1 / \widetilde{A}^1, s_{ref}^2 / \widetilde{A}^2, \ldots, s_{ref}^P / \widetilde{A}^P] \tag{13}$$

Based on (8) and (13), 2D reference and surveillance signal matrices can be obtained, establishing the signal model used in this study and laying the foundation for the following imaging process.

### 3.2. Effective Imaging Methods

In this subsection, effective target imaging methods are proposed for the developed system. Firstly, for the $p$-th pulse of the obtained signal matrices, the reference and surveillance signal vectors $s_{ref}^p$ and $s_{sur}^p$ are transformed to the discrete frequency domain, giving

$$\begin{cases} s_{ref}^p(f_{n_r}) = G_0^p(f_{n_r}) \exp(-j2\pi f_{n_r} n^p / f_s) \\ s_{sur}^p(f_{n_r}) = \sigma(\alpha, \beta) G_0^p(f_{n_r}) \exp(-j2\pi f_{n_r} n^p / f_s) \exp\{j2\pi f_{n_r}[\tau_{ref}^p - \tau_{sat}^p(\alpha, \beta) - \tau_{sur}(\alpha, \beta)]\} \end{cases} \tag{14}$$

where $f_{n_r} = f_c + [n_r - 1 - (N_r - 1)/2] f_s / N_r$ denotes the $n_r$-th frequency ($n_r = 1, 2, \ldots, N_r$), $f_c$ denotes the center frequency, and $G_0^p(f_{n_r})$ denotes the spectrum of the transmitted signal $g_0^p(\tau)$.

Then, a signal vector in the discrete frequency domain can be obtained by combing the reference and surveillance vectors, as

$$s^p(f_{n_r}) = s_{sur}^p(f_{n_r})[s_{ref}^p(f_{n_r})]^* = \sigma(\alpha, \beta) \exp\{-j2\pi f_{n_r}[\tau_{sat}^p(\alpha, \beta) - \tau_{ref}^p + \tau_{sur}(\alpha, \beta)]\} \tag{15}$$

where $[\cdot]^*$ denotes conjugate and, since the signal spectrum $G_0^p(f_{n_r})$ is flat, we set $\left| G_0^p(f_{n_r}) \right|_2 = 1$ for simplification in (15).

As the satellite is far from the antennas and the targets, the following approximations can be used based on the second-order Taylor expansions

$$\tau_{ref}^p \simeq [-\beta_{sat} - (\alpha_{sat}^p)^2 / (2\beta_{sat})] / c, \ \tau_{sat}^p(\alpha, \beta) \simeq [(\beta - \beta_{sat}) - (\alpha_{sat}^p - \alpha)^2 / (2\beta_{sat} - 2\beta)] / c \tag{16}$$

and thus

$$\tau_{sat}^p(\alpha, \beta) - \tau_{ref}^p \simeq [\beta + (\alpha_{sat}^p \alpha - \alpha^2 / 2) / (\beta_{sat} - \beta)] / c \tag{17}$$

where the term related to the square of $\alpha_{sat}^p$ is ignored considering it is relatively much smaller.

According to (17), the signal vector given in (15) can be approximated by

$$s^p(f_{n_r}) \simeq \sigma(\alpha, \beta) \exp[-j2\pi f_{n_r}(\beta_I / c + \alpha_{sat}^p \alpha_I / c)] \tag{18}$$

where $\beta_I = (\alpha^2 + \beta^2)^{1/2} + \beta - \alpha^2/(\beta_{sat} - \beta)/2$ and $\alpha_I = \alpha/(\beta_{sat} - \beta)$.

Furthermore, as the signal bandwidth is much smaller than the signal center frequency, the following approximation can be used

$$\exp\left\{-j2\pi f_{n_r}\alpha_{sat}^p\alpha/c\right\} \simeq \exp\left\{-j2\pi\alpha_{sat}^p\alpha_I/\lambda_c\right\} \tag{19}$$

where $\lambda_c$ denotes the signal wavelength corresponding to the center frequency $f_c$.

Therefore, (18) can be approximated by

$$s^p(f_{n_r}) \simeq \sigma(\alpha, \beta)\exp[-j2\pi f_{n_r}\beta_I/c]\exp[-j2\pi\alpha_{sat}^p\alpha_I/\lambda_c] \tag{20}$$

For a local imaging scene, the phase difference between (20) and (15) is always small. For example, given $\beta_{sat} = -1000$ km, satellite moving length as 1350 m (corresponding to $P = 301$ and $d = 4.5$ m, longer than those used in the field experiments), center satellite position as $\alpha_{sat}^c = 0$, and an imaging scene with $\alpha \in [-5, 5]$ km and $\beta \in [0, 10]$ km, which is the maximal imaging size in the various experiments conducted in different locations by the developed system, the maximal phase difference between (20) and (15) for all satellite positions and all frequencies of different targets in the imaging scene is shown in Figure 7a. Moreover, as the center satellite position deviates from 0 if the IW (EW) signal fragment is not ideally selected, Figure 7b shows the maximal phase difference between (20) and (15) of all the targets in the imaging scene with respect to $\alpha_{sat}^c \in [-100, 100]$ m. It can be learned from Figure 7 that, for the given parameters, the phase difference between (20) and (15) is always smaller than $\pi/2$, verifying the feasibility of applying (20) for the following target imaging process [26,27].

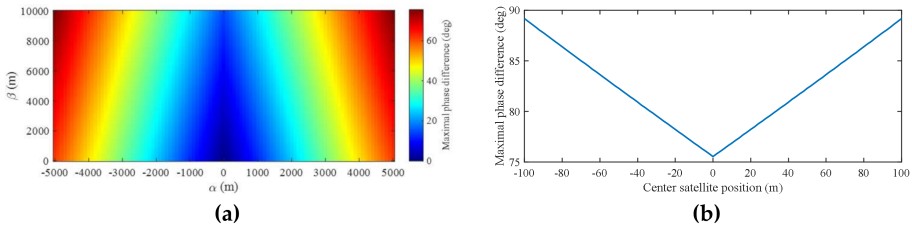

**Figure 7.** (**a**) Maximal phase difference for all satellite positions and all frequencies of different targets in the imaging scene with the satellite center position as 0 and (**b**) maximal phase difference of the imaging scene with different satellite center positions.

It can be observed from (20) that its exponential terms form the kernels of a 2D Fourier transform (FT) with respect to $\alpha_I$ and $\beta_I$, therefore, 2D inverse fast FT (IFFT) can be used to obtain a focused image as

$$\boldsymbol{\Sigma}_{LR} = \boldsymbol{F}_1^H \boldsymbol{S} \boldsymbol{F}_2^* \in C^{O \times L} \tag{21}$$

where $\boldsymbol{S} = [\boldsymbol{s}^1, \boldsymbol{s}^2, \dots, \boldsymbol{s}^P] \in C^{N_r \times P}$, $\boldsymbol{F}_1 \in C^{N_r \times O}$ and $\boldsymbol{F}_2 \in C^{P \times L}$ denote the FT matrices, given by

$$\boldsymbol{F}_1 = \exp[-j2\pi\boldsymbol{f}_r\boldsymbol{\beta}_I^T/c], \boldsymbol{F}_2 = \exp[-j2\pi\boldsymbol{\alpha}_{sat}\boldsymbol{\alpha}_I^T/\lambda_c] \tag{22}$$

with $\boldsymbol{f}_r = [f_1, f_2, \dots, f_{N_r}]^T - f_c$, $\boldsymbol{\alpha}_{sat} = [\alpha_{sat}^1, \alpha_{sat}^2, \dots, \alpha_{sat}^P]^T - \alpha_{sat}^c$, $\boldsymbol{\beta}_I \in R^{1 \times O}$ and $\boldsymbol{\alpha}_I \in C^{1 \times L}$ set according to the imaging scene size.

Based on (20), it can be derived that the $[o, l]$-th element of $\boldsymbol{\Sigma}_{LR}$ can be expressed as

$$\boldsymbol{\Sigma}_{LR}(\beta_I^o, \alpha_I^l) = \sigma(\alpha, \beta)e^{-j2\pi(\beta_I - \beta_I^o)/\lambda_c}e^{-j2\pi\alpha_{sat}^c(\alpha_I - \alpha_I^l)/\lambda_c}\operatorname{sinc}[B_r(\beta_I - \beta_I^o)/c]\operatorname{sinc}[B_a(\alpha_I - \alpha_I^l)/\lambda_c] \tag{23}$$

where $B_r = f_s$ and $B_a = Pd$ denote the signal bandwidths in the $\beta_I$ and $\alpha_I$ directions, respectively.

By formulating the imaging process as Fourier transform, the computational cost can be much reduced compared to the typical BP algorithm, and no exact satellite information is needed. However, in spite of its efficiency, the imaging process based 2D IFFT has the

same problem with the BP algorithm, i.e., the imaging resolution is limited and high-level imaging sidelobes are generated. To suppress sidelobes and improve resolution, a CS-based imaging method is proposed by exploiting the sparsity of the scene [28,29]. For the proposed method, the following minimization problem is established to achieve a high-resolution target image

$$\boldsymbol{\Sigma}_{HR} \leftarrow \min_{\boldsymbol{\Sigma}} \frac{1}{2} \left| \left| \boldsymbol{S} - \boldsymbol{F}_1 \boldsymbol{\Sigma} \boldsymbol{F}_2^T \right| \right|_F^2 + \xi \left| \left| \boldsymbol{\Sigma} \right| \right|_1 \tag{24}$$

where $||\cdot||_F$ denotes the Frobenius norm, $||\cdot||_1$ denotes the $L_1$ norm, representing the sparsity of $\boldsymbol{\Sigma}$, and $\xi$ denotes the regularization parameter.

An effective way to solve (24) is to use the 2D fast iterative soft thresholding algorithm (FISTA) [30], whose $[k+1]$-th iteration can be expressed as

$$\begin{cases} \boldsymbol{\Sigma}_{k+1} = \text{soft} \left[ \boldsymbol{Z}_k + \mu \boldsymbol{F}_1^{\text{H}} (\boldsymbol{S} - \boldsymbol{F}_1 \boldsymbol{Z}_k \boldsymbol{F}_2^T) \boldsymbol{F}_2^*, \theta \right] \\ \eta_{k+1} = (1 + \sqrt{1 + 4\eta_k^2})/2 \\ \boldsymbol{Z}_{k+1} = \boldsymbol{\Sigma}_{k+1} + (\eta_k - 1)(\boldsymbol{\Sigma}_{k+1} - \boldsymbol{\Sigma}_k)/\eta_{k+1} \end{cases} \tag{25}$$

where soft $[x, \theta] = x/|x|\max(|x| - \theta, 0)$ denotes the soft thresholding function with $\theta$ as the threshold, $\eta_{k+1}$ denotes a variable, and $\mu \in (0, 1/||\boldsymbol{F}_1||_F^2||\boldsymbol{F}_2||_F^2)$ denotes the step size.

In summary, the high-resolution imaging algorithm based on 2D FISTA for the developed SDR MF-PB-SAR system is shown in Algorithm 1.

---

**Algorithm 1** The proposed high-resolution imaging algorithm based on 2D FISTA

---

**Input:** $\boldsymbol{S}$, $\boldsymbol{F}_1$, $\boldsymbol{F}_2$, $\mu$, $\theta$, the maximal iteration number $K$, and the stop parameter $\varsigma$.
**Initial:** $\boldsymbol{\Sigma}_0 = 0$, $\boldsymbol{Z}_0 = 0$, and $\eta_0 = 1$.
**for** $k = 0$ **to** $K - 1$ **do**
    $\boldsymbol{J}_k = \boldsymbol{F}_1^{\text{H}} (\boldsymbol{S} - \boldsymbol{F}_1 \boldsymbol{Z}_k \boldsymbol{F}_2^T) \boldsymbol{F}_2^*$;
    $\boldsymbol{\Sigma}_{k+1} = \text{soft} [\boldsymbol{Z}_k + \mu \boldsymbol{J}_k, \theta]$;
    $\eta_{k+1} = (1 + \sqrt{1 + 4\eta_k^2})/2$;
    $\boldsymbol{Z}_{k+1} = \boldsymbol{\Sigma}_{k+1} + (\eta_k - 1)(\boldsymbol{\Sigma}_{k+1} - \boldsymbol{\Sigma}_k)/\eta_{k+1}$;
  **if** $||\boldsymbol{\Sigma}_{k+1} - \boldsymbol{\Sigma}_k||_F/||\boldsymbol{\Sigma}_k||_F < \varsigma$ **then**
    $K_0 = k + 1$;
    stop iteration.
  **end if**
**end for**
**Return:** $\boldsymbol{\Sigma}_{HR} = \boldsymbol{\Sigma}_K$ **or** $\boldsymbol{\Sigma}_{HR} = \boldsymbol{\Sigma}_{K_0}$.

---

Based on $\boldsymbol{\Sigma}_{LR}$ or $\boldsymbol{\Sigma}_{HR}$, the reflection coefficient of the target at $(\alpha, \beta)$ can be obtained by the interpolation process, expressed as

$$\widetilde{\sigma}(\alpha, \beta) = \Sigma[\alpha/(\beta_{sat} - \beta), (\alpha^2 + \beta^2)^{1/2} + \beta - \alpha^2/(\beta_{sat} - \beta)/2] \tag{26}$$

which forms the imaging method on the $\alpha$-$o$-$\beta$ plane.

At last, in order to achieve the target image on the $x$-$o$-$y$ plane, the following approximated relationships of $(x, y)$ and $(\alpha, \beta)$ are applied in this study:

$$\alpha = y, \tau_{sat}^z(\alpha, \beta) + \tau_{sur}(\alpha, \beta) = \tau_{sat}^z(x, y) + \tau_{sur}(x, y) \tag{27}$$

where the superscript 'z' denotes that the satellite is at the position with $\alpha_{sat} = y_{sat} = 0$, based on which the second equation in (27) can be simplified to

$$\sqrt{y^2 + (H/\sin\varphi + \beta)^2} + \sqrt{y^2 + \beta^2} = \sqrt{(H\cot\varphi + x)^2 + y^2 + H^2} + \sqrt{x^2 + y^2} \tag{28}$$

Therefore, given the target at $(x, y)$ on the ground, its corresponding point $(\alpha, \beta)$ on the $\alpha$-$o$-$\beta$ plane can be obtained based on the first equation in (27) and the solution of (28),

providing the image on the *x-o-y* plane achieved by interpolation according to the process in (26).

## 4. Experiment Results

Thanks to the transportable capability of the developed SDR MF-PB-SAR system, various experiments were conducted at different locations. In this section, some imaging results are presented to validate the system and the proposed methods.

The first experiment we conducted is at a mountainous area in Besançon, France, as shown in Figure 8, where the aerial picture of the scene is also shown as the reference for imaging, provided by OpenStreetMap. The observation is during the ascending pass of Sentinel-1 so that the scene is illuminated from the west and the surveillance antenna is oriented to the east to collect the signals reflected by the targets. Here, the IW 3 signal is selected and processed, where the incident angle is set as $\varphi = 43°$ and all other parameters are the same as Figure 5a, i.e., $\chi = 60$, $Nr = 2401$, $P = 251$, and $T = 593.18$ µs. It should be noted that the dataset used to draw Figure 2, Figure 3, and Figure 5 are obtained by this experiment.

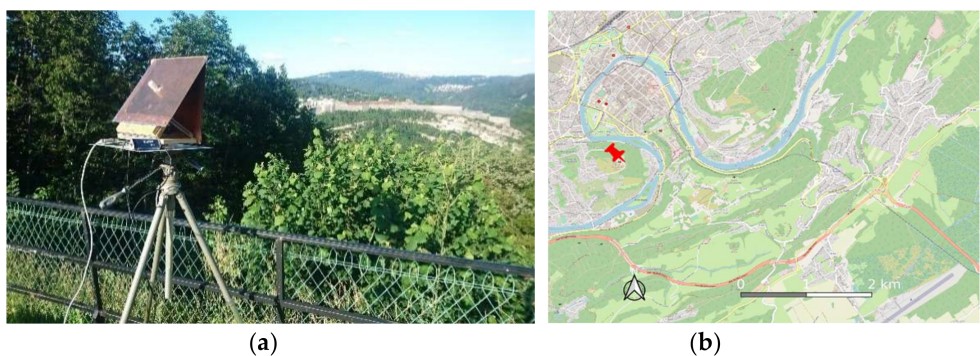

(a)          (b)

**Figure 8.** (**a**) Setup of the experiment conducted in Besançon, France, at the Fort Chaudanne location and (**b**) the aerial picture of the observed scene with the experiment site indicated by the red nail.

The imaging results with respect to $\alpha_I$ and $\beta_I$ obtained by 2D IFFT-based method and 2D FISTA-based method are shown in Figure 9 with a 40 dB dynamic range. It can be seen that both methods can achieve well-focused images of the scene and display some clear features, while the sparsity-based method enjoys higher imaging resolution and low sidelobe level, which is more clearly shown in Figure 10. In Figure 10, two cuts along the abscissa of Figure 8a,b for an ordinate with two isolated targets are compared, demonstrating the sidelobe reduction and resolution improvement achieved by the 2D FISTA-based imaging method.

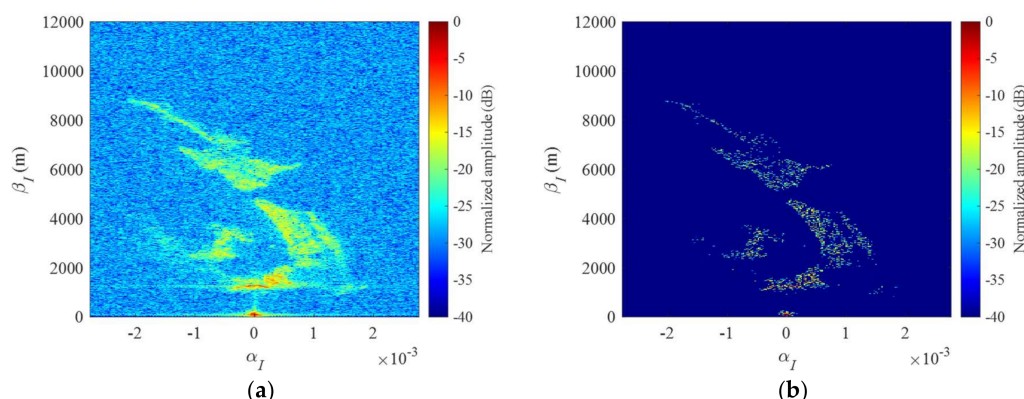

(a)          (b)

**Figure 9.** Imaging results of the mountainous area in Besançon obtained by (**a**) 2D IFFT and (**b**) 2D FISTA; the abscissa and ordinate are $\alpha_I$ and $\beta_I$, respectively.

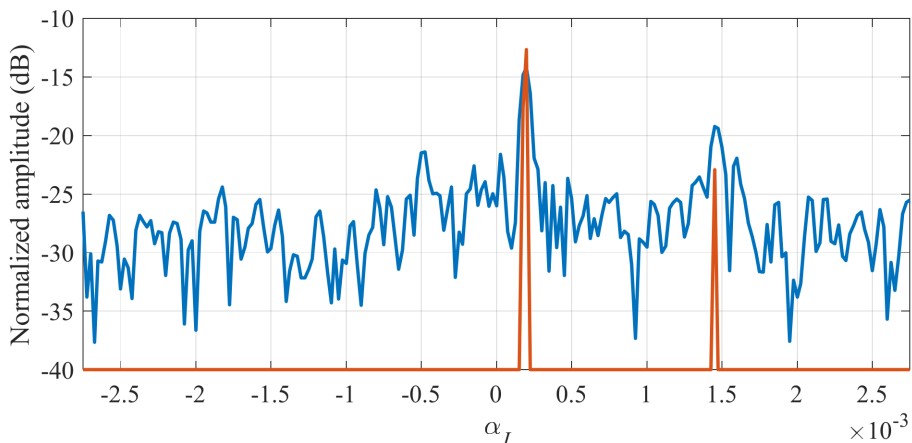

**Figure 10.** Sidelobe reduction and resolution improvement as observed from the cross-sections of Figure 9a,b for an ordinate with two isolated targets.

To show the functions of DPI suppression as that in (8) and amplitude compensation as that in (13), the imaging results obtained by 2D IFFT without DPI suppression and by 2D FISTA without amplitude compensation are shown in Figure 11. It can be observed by comparing Figures 9a and 11a that the DPI has negative influence on target imaging. As indicated by the red rectangle in Figure 11a, some artifacts are generated by the DPI. It can also be observed by comparing Figures 9b and 11b that, as the signal obtained without amplitude compensation is not exactly consistent with the model established in (20), although the influence is not obvious, the imaging result has more artifacts than that obtained with amplitude compensation.

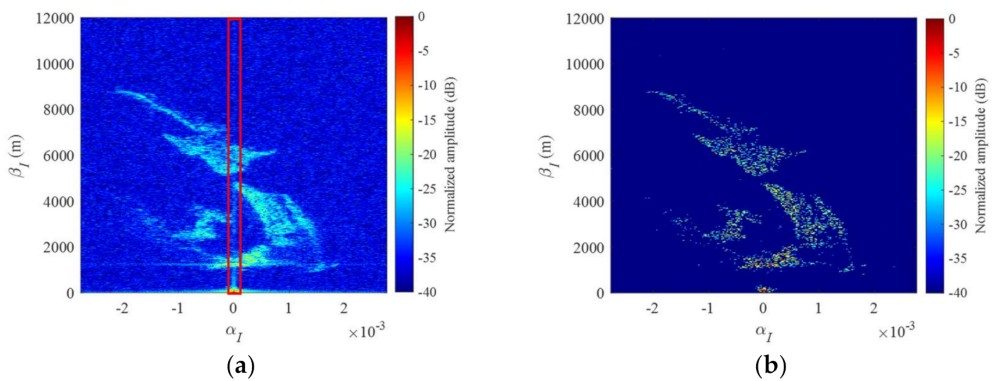

(**a**)                                          (**b**)

**Figure 11.** Imaging results of the mountainous area in Besançon obtained by (**a**) 2D IFFT without DPI suppression as that in (8) and (**b**) 2D FISTA without amplitude compensation as that in (13).

Based on the image obtained by 2D FISTA as shown in Figure 9b, the MF-PB-SAR images on the $\alpha$-$o$-$\beta$ plane (the scene size is included in Figure 7a, hence the feasibility of signal approximations) and on $x$-$o$-$y$ plane are obtained by solving (26) and using the linear interpolation method, as shown in Figure 12. It can be observed that the abscissa/ordinate of Figure 12 are magnified/minified as compared to Figure 9b, matching the ground scale of the actual scene. To assess the imaging performance, the obtained MF-PB-SAR image on the $x$-$o$-$y$ plane is overlaid above the aerial picture and the digital elevation model (DEM), as illuminated in Figure 13, where the abscissa of the obtained image is rotated to be parallel to the satellite flight path. It can be seen that the obtained image matches well with the real scene: the mountains, man-made structures in the river shores (as indicated by rectangles), and cliff (as indicated by the arrow) can be identified.

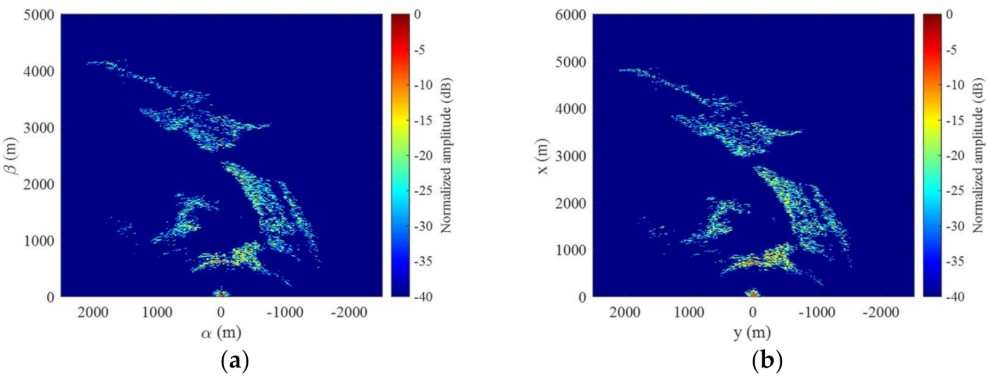

(**a**)  (**b**)

**Figure 12.** MF-PB-SAR images of the mountainous area in Besançon on (**a**) the $\alpha$-$o$-$\beta$ plane and (**b**) the $x$-$o$-$y$ plane, obtained by the linear interpolation process.

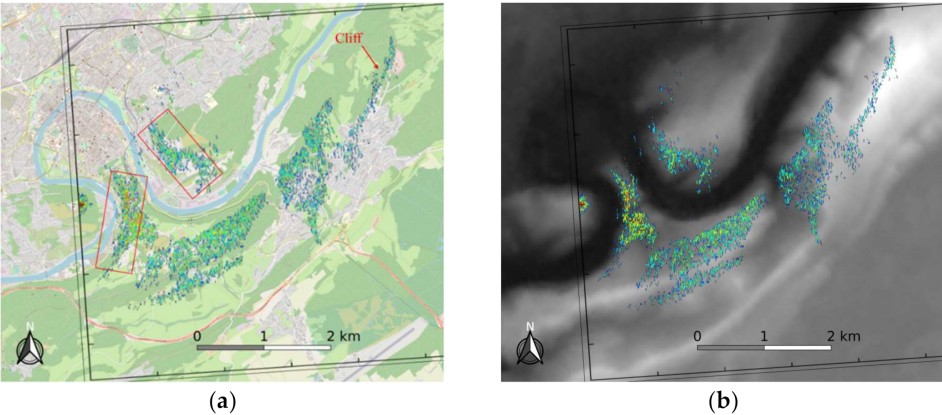

(**a**)  (**b**)

**Figure 13.** MF-PB-SAR image of the mountainous area in Besançon during an ascending pass with the satellite beam facing east, overlaid with (**a**) the aerial picture provided by OpenStreetMap and (**b**) the digital elevation model (DEM).

The second experiment we conducted is at an urban area in Paris, France, as shown in Figure 14. This aera contains abundant targets, e.g., building, street, and park, for imaging and the landmark of Paris, Eiffel Tower, is about 5.73 km from the experiment site, as indicated by the red star in Figure 14b. During the measurement, the ascending pass of Sentinel-1 illuminates the city from the west, so that the surveillance antenna is facing eastward. The processing parameters are: $\varphi = 52°$, $\chi = 60$, $Nr = 2401$, $P = 275$, and $T = 688.88$ µs (i.e., the IW 2 signal fragment is selected in this experiment).

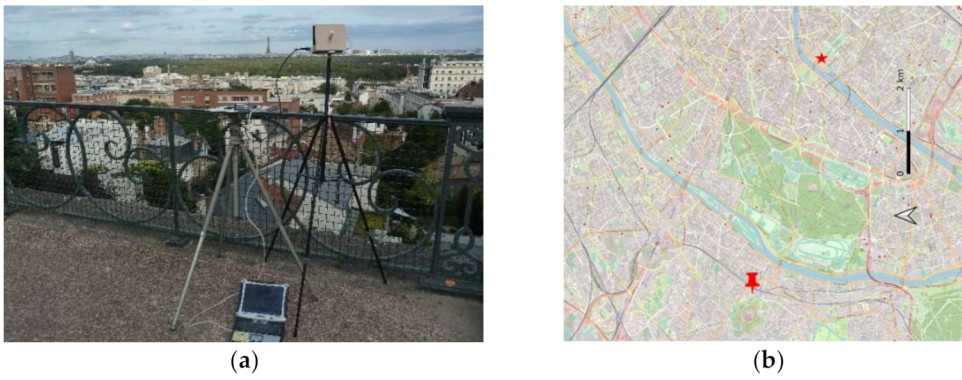

(**a**)  (**b**)

**Figure 14.** (**a**) Setup of the experiment conducted in Paris, France, at the Mont Valerien location and (**b**) the aerial picture of the imaging scene, where the experiment site is indicated by the red nail.

Figure 15a,b show the imaging results obtained by 2D FISTA on the $\alpha$-$o$-$\beta$ plane (the scene size is still included in Figure 7) and the $x$-$o$-$y$ plane, respectively. It can be seen that the images are well focused, and different ground features can be easily observed. By overlaying the MF-PB-SAR image on the $x$-$o$-$y$ plane with the aerial picture provided by OpenStreetMap, as shown in Figure 16, the imaging result can be verified: long street (indicated by the rectangle), tall buildings (indicated by the circles), river-shore man-made structure, park (indicated by the polygon), and the Eiffel Tower match well with the actual scene.

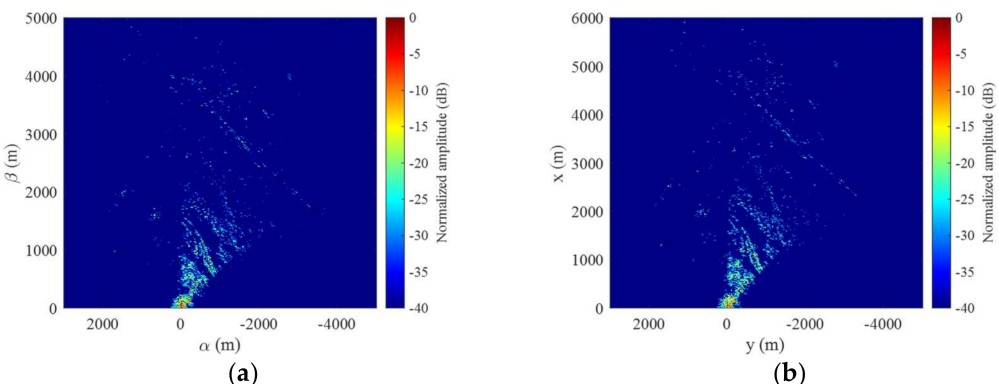

**Figure 15.** MF-PB-SAR images of the urban area in Paris (**a**) on the $\alpha$-$o$-$\beta$ plane and (**b**) on the $x$-$o$-$y$ plane, obtained by the 2D FISTA-based imaging method.

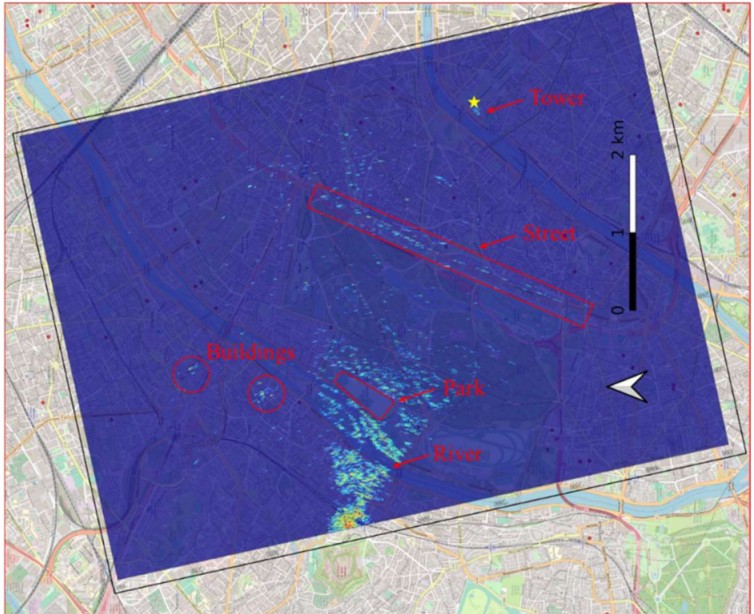

**Figure 16.** MF-PB-SAR image of the urban area in Paris, overlaid with the aerial picture.

The third experiment we conducted is at Spitsbergen, Norway, close to the North Pole, as shown in Figure 17. Different from previous two experiments, EW illumination signal with a PRI of $T = 613.25$ μs are sampled and used for target imaging in this case. Other processing parameters are: $\varphi = 45°$, $\chi = 60$, $N_r = 3201$ (to achieve a longer detection range than $N_r = 2401$), and $P = 201$. Moreover, to verify the stability and repeatability of the developed system, two datasets were sampled with an interval of 12 days (i.e., the period of the satellite) at 22 September and 4 October 2021.

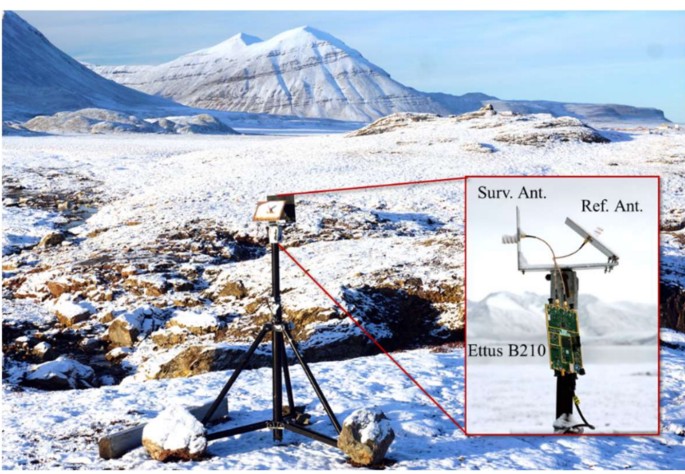

**Figure 17.** Setup of the experiment conducted in Spitsbergen, Norway. The system is fixed on the same position for a period of 12 days to show its stability and repeatability.

The imaging results at two different dates obtained by the 2D FISTA-based imaging method on the $\alpha$-$o$-$\beta$ plane (the scene size is still included in Figure 7) and the $x$-$o$-$y$ plane are shown in Figure 18. It can be seen that multiple snow mountains can be well imaged and the imaging results corresponding to two different dates are similar to each other, demonstrating the stability of the developed system. Figure 19 shows the overlay between the MF-PB-SAR image (colorful) and the SAR image (grey) generated by Sentinel-1 during a pass at 3 October. It can be learned that the match is reasonable, and the obtained image can show different scattering properties of the mountains from a bistatic angle. It should be mentioned that, although the measurements at two different dates allow for phase comparison in the context of differential interferometry process, detailed analysis is ongoing.

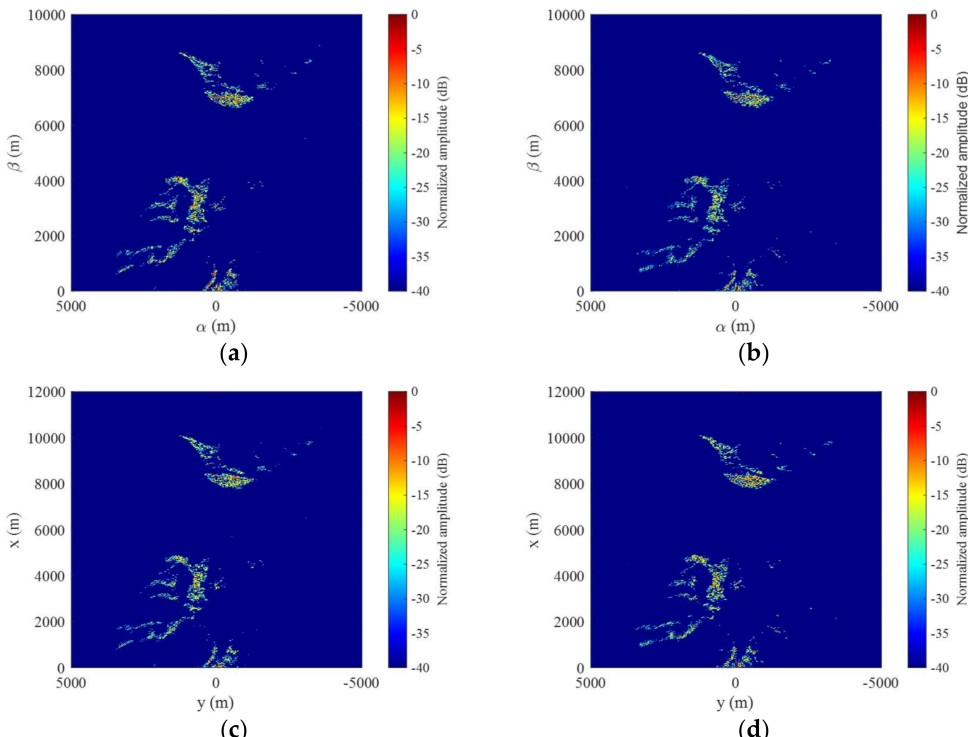

**Figure 18.** MF-PB-SAR images of the snow mountains in Spitsbergen (**a**,**b**) on the $\alpha$-$o$-$\beta$ plane and (**c**,**d**) on the $x$-$o$-$y$ plane. The left sub-figures and right sub-figures correspond to the data sampled in 22 September and 4 October, respectively.

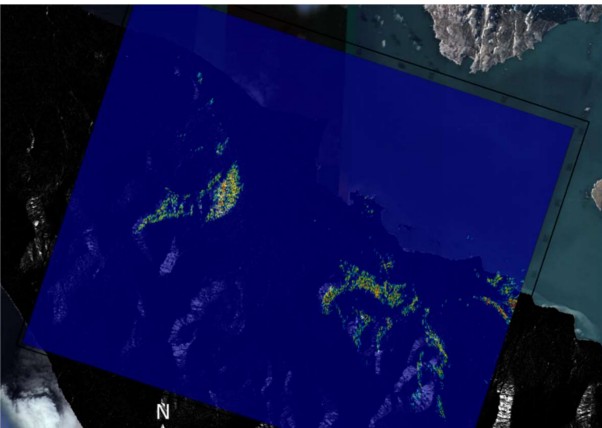

**Figure 19.** Overlay between the MF-PB-SAR image obtained in Spitsbergen and the SAR image generated by Sentinel-1.

## 5. Conclusions

By using commercial-off-the-shelf software-defined radio hardware, a passive bistatic synthetic aperture radar using the C-band SAR satellite, Sentinel-1, as its emitting source has been developed and validated, with no need for accurate time and position synchronization between the satellite in space and the fixed receiver on the ground. Its system structure, signal model, imaging geometry, signal pre-processing method, and approximation-based effective target imaging methods have been introduced in this paper. Experiments in mountainous and urban areas have been conducted, and the imaging results have shown that the developed system and the proposed methods can obtain well-focused images that match the actual local scene in a range of several kilometers. Running the full data acquisition on a Raspberry Pi 4 single-board computer makes the system deployable for autonomous remote sensing measurements. Data acquisition and processing software is available at https://github.com/jmfriedt/sentinel1_pbr (accessed on 1 November 2021). In the future, high-resolution target imaging methods that combine multiple signal fragments and differential interferometry processing methods will be studied, and the application of the developed MF-PB-SAR system, such as the displacement monitoring of mountains, buildings, and towers, will be carried out.

**Author Contributions:** J.-M.F. proposed the idea, conducted the experiments, analyzed the data, and helped to write the manuscript; W.F. established the signal model, proposed the processing methods, and wrote the manuscript; P.W. helped to write the manuscript and gave suggestions on the processing methods. All authors have read and agreed to the published version of the manuscript.

**Funding:** This research was funded by National Natural Science Foundation of China under grant number 62001507 and Young Talent fund of University Association for Science and Technology in Shaanxi, China under grant number 20210106. The APC and the SDR hardware were funded by the French Space Agency CNES under grant number R-S18/LN-0001-036. The trip to Spitsbergen was funded by the French Paul Emile Victor Polar Institute IPEV as part of the PRISM grant.

**Institutional Review Board Statement:** Not applicable.

**Informed Consent Statement:** Not applicable.

**Data Availability Statement:** https://github.com/jmfriedt/sentinel1_pbr (accessed on 1 November 2021).

**Acknowledgments:** The antennas were fabricated at the mechanical workshop of FEMTO-ST (Besançon, France) by P. Abbe and V. Tissot.

**Conflicts of Interest:** The authors declare no conflict of interest.

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
