# Peer review of "SDR-Implemented Passive Bistatic SAR System Using Sentinel-1 Signal and Its Experiment Results"

_remotesensing, doi:10.3390/rs14010221_

Round 1

Reviewer 1 Report

the idea of a passive bistatic synthetic aperture radar is not new but the usage of a commercial off the shelf card to acquire and process the signal is a the idea exposed in this publication. On this view, the article is original. It expose clearly the method used to obtened results and the data in order to evaluate their quality compared to a standard design of such kind of radar.

On lines 183, 192, replace sref by the right parameter: ssur. On page 12, figure 12, if possible improve the transparency of radar map over the the aerial picture for a better reading.

Author Response

Please find attached in the PDF document the detailed response to each reviewer comments. Thank you.

Reviewer 2 Report

  1. Line 180, the symbols of the reference signal and surveillance signal are the same, which is needed to be corrected.
  2. The definition of the reference signal and surveillance signal is not clear, which is difficult for readers to understand the signal and geometry model for the proposed algorithm. It is suggested to give a certain illustration of the signal.
  3. Line 228, The description of coordinate is very confusing. The trajectory of the satellite was not mentioned before, and the center position can not be set without the trajectory. It is recommended to add to this part.
  4. The signal history is not shown in Fig.5. For example, where is surveillance signal or reference It is hard for readers to find the relationship between the geometry model and the signal model, which is suggested to be modified
  5. Line 252-253, the logical of the sentence “Since it is always much stronger than the target reflections, the DPI component in will have seriously negative influence on the imaging performance may be wrong.
  6. Line 265, the definition of A is shown in the paper. Readers may not understand the reason why the A is needed to estimate.
  7. The mathematical symbols used in section 3 are confusing. Although there are a sufficient number of formulas in section 3.2, the main work for your study is not well reflected from the imaging method. It is suggested to Highlight the novelty of the algorithm by formula rather than only describing the process procedure.
  8. Experiment results in section 4 are not sufficient. Some experiments compared with other methods are needed to prove the performance of the algorithm.

Author Response

(The authors gave the same response as above.)

Reviewer 3 Report

Dear authors,

it was a pleasure to review this well written paper. What you describe is a great piece of engineering. I very much liked the pragmatism that you used for determining the flight details of the satellite and its timing, both described on page 5.

Please consider the following remarks.

1) in line 182 and line 183 the text states sref of which one surely is meant to be ssur.

2) line 236 and many other occasions. The term PRI is used in a wrong way. The PRI is a single number, measured in microseconds. Please do not use the same term with a different meaning like "pth PRI". That should be something like "pth pulse". This problem becomes severely difficult when the PRI is used as the label for the x-axis in Fig.5 but the caption defines the PRI being three different fixed values.

 3) line 24. I strongly disagree with this statement. For example, PBR requires much more expenses than FMCW radars which are extremely cheap COTS components of systems like autonomous cars.

Author Response

Please find in the attached PDF document the detailed response to each reviewer comments. Thank you.

Reviewer 4 Report

This article introduces a passive bistatic SAR system. The experimental results show that the hardware and software are running well. The results are very  impressive.

Author Response

(The authors gave the same response as above.)
